# Thomas Aquinas on Gratitude to God

Christopher Kaczor 

Department of Philosophy, Loyola Marymount University, Los Angeles, CA 90045, USA; ckaczor@lmu.edu

**Abstract:** Discussions of gratitude to God characteristically presuppose some philosophical or theological framework. This philosophical and theological exploration of gratitude to God examines the topic in light of the thought of Thomas Aquinas. Unlike some treatments of Aquinas' account of gratitude, I draw extensively on Aquinas' commentaries on Scripture as well as lesser known works, such as his sermons, to illuminate these topics rather than exclusively relying on the *Summa theologiae*. In the first part of this article, I focus on how Aquinas understands the virtue of gratitude to God. In the second part, I examine his account of ingratitude to God. And in the third part, I consider the difference Jesus makes in Aquinas' understanding of these issues, including contesting the claim that "Jesus was an ingrate".

**Keywords:** Aquinas; virtue; religion; gratitude; gratitude to God



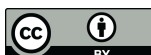

## 1. Introduction

What is gratitude? Aquinas defines the term as "recollecting the friendship and kindliness shown by others, and in desiring to pay them back, as Tully states (*De invent*. ii, 53)." (Aquinas 1920, II-II, 80, 1). Various kinds of benefactors can show us friendship and kindness. Aquinas notes that "corresponding to these various kinds of debt there are various virtues: e.g., *Religion* whereby we pay our debt to God; *Piety*, whereby we pay our debt to our parents or to our country; *Gratitude*, whereby we pay our debt to our benefactors, and so forth" (Aquinas 1920, I-II, 60, 3). Even though, strictly speaking, this way of categorizing various debts would lead to gratitude *not* being due to God or to parents (since religion and piety correspond to these debts), Aquinas does indeed sometimes speak of gratitude to God and explores the topic in a variety of works. In this essay, I gather the obiter dicta of Aquinas on gratitude to God, on the various forms of ingratitude to God, and on how Jesus makes a difference for gratitude to God. I also call into question the claim that "Jesus was an ingrate" (Leithart 2014, p. 68).

## 2. Gratitude to God

In his consideration of religion as expressed in Jewish liturgy, Aquinas asks whether there was a suitable order for the liturgical rites. He affirmatively answers, "The reason for this order is that man is bound to God, chiefly on account of His majesty; secondly, on account of the sins he has committed; thirdly, because of the benefits he has already received from Him; fourthly, by reason of the benefits he hopes to receive from Him." (Aquinas 1920, I-II, 102, 3 ad 10). Gratitude to God is part of due worship of God, but in third place. The highest form of liturgy is adoration of God for his Perfect excellence, the next highest is contrition for sin against God, the third is thanksgiving for God's blessings, and last is the prayer of petition asking for future blessings. Gratitude to God is not, in his view, the most important aspect of the virtue of religion.

Of course, we cannot "pay back" our debts to some people. We cannot give birth to our parents; nor can we bring the Uncaused Cause into existence. Taking this inability to repay into account, Aquinas writes:

> In repaying favors we must consider the disposition rather than the deed. Accordingly, if we consider the effect of beneficence, which a son receives from his

parents namely, to be and to live, the son cannot make an equal repayment, as the Philosopher states (Ethic. viii, 14). But if we consider the will of the giver and of the repayer, then it is possible for the son to pay back something greater to his father, as Seneca declares (De Benef. ii). If, however, he were unable to do so, the will to pay back would be sufficient for gratitude. (Aquinas 1920, II-II, 106, 6, ad 1)

So, gratitude is a matter of disposition rather than a deed. This distinguishes gratitude from justice in typical cases. The just act must be accomplished, rather than simply willed. By contrast, Aquinas thinks of gratitude as primarily a matter of will, rather than of external deeds of repayment. He writes, "No man is excused from ingratitude through inability to repay, for the very reason that the mere will suffices for the repayment of the debt of gratitude, as stated above (Q. 106, A. 6, ad 1)" (Aquinas 1920, II-II, 107, 1, ad 2). Gratitude is more a matter of the interior than the exterior, and more a matter of the heart eager to repay than the deed of repaying.

In this way, gratitude to God as found in the virtue of religion is also to be distinguished from justice, properly speaking in that gratitude, unlike justice in the usual sense, involves the passions, whereas justice in itself is about operations (Aquinas 1920, II-II, 58, 9). If I owe you money, and I pay you back what I owe, then I have done an act of justice. Gratitude to God, by contrast, is more a matter of the heart, of the emotional life, of desire. Aquinas writes,

A poor man is certainly not ungrateful if he does what he can. For since kindness depends on the heart rather than on the deed, so too gratitude depends chiefly on the heart. Hence Seneca says (De Benef. ii): "Who receives a favor gratefully, has already begun to pay it back: and that we are grateful for favors received should be shown by the outpourings of the heart, not only in his hearing but everywhere.'" (Aquinas 1920, II-II, 106, 3, ad 5)

In giving a favor, the affections of the heart and not just the gift are relevant, so too with repaying the favor. Aquinas notes that "Seneca says (De Benef. ii): 'Do you wish to repay a favor? Receive it graciously'" (Aquinas 1920, II-II, 106, 4). Aquinas believes we cannot pay back God what we've been given. But in such cases, paying it back is done by "outpourings of the heart" not only directly to the giver but also to others who learn of the generosity of the giver. In the case of God, this is to give glory to God, not as if in itself God's glory (in the sense of intrinsic excellence) is augmented by praise but because the excellence of God can be made better known among human beings by way of our public praise (Stump 2012, p. 329). Religion as not merely a private but a social virtue involves public adoration, repentance, thanksgiving, and petition to God.

The importance of the heart is also emphasized by Aquinas in his *Commentary on the Psalms*. Aquinas writes, "Thanks is given in three ways: in heart, in words, and in deed." (Aquinas 2020a). Aquinas continues explicating the Psalm: "*I will be glad*. Here he gives thanks in the heart." (Aquinas 2020a). Rejoicing is a way of giving thanks to God in the heart which, for Aquinas, can be considered in two respects:

First, when we rejoice in the Divine good considered in itself; secondly, when we rejoice in the Divine good as participated by us. The former joy is the better, and proceeds from charity chiefly: while the latter joy proceeds from hope also, whereby we look forward to enjoy the Divine good, although this enjoyment itself, whether perfect or imperfect, is obtained according to the measure of one's charity. (Aquinas 1920, II-II, 28, 1)

Aquinas writes first of gratitude to God in the sense of rejoicing in who God is. To recognize and adore Perfect Goodness, Truth, and Beauty is to rejoice in God. Secondarily, however, gratitude to God is also due for gifts God has given or will give to us. The order of importance is not accidental. To do otherwise would be to focus on what is given rather than on the giver. In as much as Aquinas believes God is Absolute Goodness, to value the

gifts of God more than God himself is to act irrationally and wrongly by valuing the lesser good over the greater good.

Not only is God the greatest benefactor, but God also supplies the greatest benefit. Aquinas understands that benefactors aid us to different degrees. When the benefactor in question is God, our debt is the greatest of all because God is the First Cause of creation. In commenting on the Apostle's creed, Aquinas writes:

> We are led to give thanks to God. Because God is the Creator of all things, it is certain that what we are and what we have is from God: "What do you have that you did not receive?" [1 Cor 4:7]. "The earth is the Lord's and the fullness thereof; the world and all who dwell on it" [Ps 23:1]. "We, therefore, must give thanks to God: What shall I render to the Lord for all the things that He has done for me?" [Ps 115:12]. (Aquinas 2020b)

So, for Aquinas, gratitude to God is part of giving to God what is due to God as our creator. Gratitude to God involves awareness that God as the First Cause gives us all the good things in our lives—friends, family, faith, and our very lives—as gifts.

Indeed, "gifts" is the right word. According to Aquinas, God did not have to create anything at all. The Divine Freedom did not have to create the world as he created it. God could have created a world without the friends we love, without the beauty we enjoy, and without the delights we experience. Our very existence is a gift. After becoming aware of what God has given to us, gratitude also enjoins us to give thanks to God for these blessings. Finally, gratitude calls us to give to God something in return. For Aquinas, part of what is due to God is worship. Aquinas' account of gratitude differs from accounts of gratitude as merely a helpful practice for psychological well-being. For Aquinas, in order to be a just person, an agent has an ethical obligation to give gratitude to God.

Gratitude to God, understood as part of what Aquinas calls the religion, is a virtue. What then is virtue? Thomas Aquinas distinguishes two kinds of virtue: the infused and the acquired. The infused virtues are a free gift from God. The acquired virtues are gained through our own repeated actions. In defining what an infused virtue is, Aquinas endorses the definition he finds in Augustine of Hippo: "Virtue is a good quality of the mind, by which we live righteously, of which no one can make bad use, which God works in us, without us" (Aquinas 1920, I-II, q.55, a.4). Virtue is a habitual quality of the mind, but not just any quality of mind. The human mind may work for a variety of ends (e.g., financial, social, material). Good qualities of mind include mathematical understanding, ability in writing poetry, and expertise in legal matters. The phrase "by which we live righteously" suggests that the quality of mind called virtue is linked to ethical aims, acting in accordance with right reason and doing the right thing. The phrase "of which no one can make bad use" indicates that virtues (unlike skills and unlike knowledge) cannot be used for immoral purposes. For example, a surgeon can use her surgical skills to heal or to harm the patient. By contrast, a just person cannot make use of justice to do injustice. Finally, a virtue is that "which God works in us, without us." By this, Augustine and Aquinas mean that this good quality is a gift from God rather than a human accomplishment. Just as no one causes himself to be alive, so too no one causes herself to have the infused virtues.

Aquinas, again following Augustine, recognize that infused virtue and acquired virtue differ in one crucial aspect. In the definition of an acquired virtue, the phrase "which God works in us, without us" is missing because an acquired virtue is one attained through human actions. If someone repeatedly does just, courageous, temperate, and practically wise actions, that person builds just, courageous, temperate, and practically wise habits. As mentioned, Aquinas holds that our very lives (and hence the capacities we have to become virtuous) are themselves gifts from God. But these acquired virtues do not require (in Aquinas' view) extra supernatural power directly from God in order to be attained.

When considering the relationship between gratitude and virtue, we can then distinguish two different questions: (1) "Is gratitude to God the foundation or prerequisite to the infused virtues?" and (2) "Is gratitude to God the foundation or prerequisite to the acquired virtues?" In considering the first question, it is important to remember that the infused

virtues are unearned gifts from God. Infused virtues include the theological virtues of faith, hope, and love as well as infused practical wisdom, infused justice, infused temperance, and infused courage. These infused virtues as free gifts of grace are not earned by us, and therefore require no action on our part, though we may reject them. As the definition of the infused virtues states, these good qualities are what "God works in us, without us". Aquinas held that even a baby, through the gift of baptism, can have the infused virtues. But an infant, due to a lack of knowledge, cannot recognize a benefit or a benefactor or her status as a beneficiary, so a baby cannot have gratitude. This point applies not only to babies, but also to adults. The theological virtues are gifts of grace for the mature adult just as much as for the infant. These gifts are not earned by our action and do not depend upon our action, including our willingness to recognize God's benefits to us. So, gratitude to God is not the foundation or prerequisite of the infused virtues.

One can then ask, "Is gratitude to God the foundation or prerequisite to the acquired virtues?" The acquired virtues are gained through our repeated actions. The temperate person becomes temperate by eating the right amount of food and in the right way, repeatedly over time. Likewise, the virtue of justice is gained by giving to each person what is due over and over again until it becomes an ingrained habit to give each person what is due to that person. Similarly, an agent gains acquired courage and practical wisdom through repeated actions of those kinds. The question then becomes: is it possible to commit a just act, a temperate act, a courageous act, or a practically wise act without gratitude to God?

It seems that it is possible for the atheist to do a just act (say, pay back a debt) without acknowledging that God exists, let alone showing gratitude to God. Similarly, it would seem to be true that the atheist could do a temperate act (say, eating the proper amount of food) without acknowledging the gifts that God has given. If this analysis is correct, then gratitude to God is not the foundation or prerequisite for having the acquired virtues.

For Aquinas, the foundation and prerequisite of all true and perfect virtue is not gratitude to God but rather love as a gift of God. True and perfect virtue is not acquired but infused. He holds, following St. Paul in First Corinthians (13:3), that without love we cannot have true and perfect virtue of any kind (Aquinas 1920, II-II, q.23, a.7). Without love, the best a person can have is true but imperfect acquired virtue, since without love the human person cannot achieve the final end of all human life, an everlasting friendship with God. Aquinas holds that love is the greatest of the virtues.

But what exactly does it mean to say that a given virtue is greater than the others? Is gratitude (to God) the greatest of the virtues? How might we judge one virtue as greater than another? One virtue is greater than another, Aquinas says, because its "object" is greater. The object of the virtue is the focus of the virtue. For Aquinas, faith, hope, and love all concern God, the greatest good, in different dimensions. Faith connects us to God as First Truth. Hope connects us to God as Source of Perfect Happiness. Love unites us in loving friendship with God. Because faith, hope, and love connect us to God, who is the greatest good, these theological virtues are greater than virtues whose objects are of lesser importance, such as facing dangers (courage), enjoying bodily pleasure in accordance with the demands of what is right (temperance), or giving to each person what is due to each person (justice). Likewise, Aquinas says the virtue of religion (giving to the Creator what is due to the Creator) is a greater virtue than other acts of justice (say, giving to parents what is due to parents). So, the more directly a virtue connects us to God, the greater that virtue.

Aquinas notes another way in which love is greater than other virtues. In heaven, faith is no longer needed because we see God face to face. In heaven, hope (which concerns a possible but not certain future good) is no longer needed because we have actually attained and currently enjoy perfect Happiness. In heaven, love remains (Aquinas 1920, II-II, 24, 8). So, it would seem, one virtue is greater than another if it is more long-lasting than another.

But in this respect, gratitude to God is a greater virtue than faith or hope, since gratitude to God remains in heaven:

> I answer that the cardinal virtues will remain in heaven, but only as regards the acts which they exercise in respect of their end. Wherefore, since the virtue of

penance is a part of justice which is a cardinal virtue, whoever has the habit of penance in this life will have it in the life to come. But he will not have the same act as now, but another, namely, thanksgiving to God for his mercy in pardoning his sins. (Aquinas 1920, IV 16, 2)

If we consider a virtue to be greater inasmuch as it is more long-lasting, then gratitude to God is greater in this respect than faith or hope, both of which no longer exist in heaven. So, according to Aquinas, in heaven we both remember our sins and give thanks to God for his mercy in forgiving our sins, which makes gratitude to God, in this respect, greater than faith and hope.

In the *Summa theologiae*, Aquinas places his discussion of gratitude within the context of justice. Yet, in the *Summa theologiae,* as well as in his commentaries on Paul, Aquinas also links gratitude to love. He writes, "The debt of gratitude flows from the debt of love, and from the latter no man should wish to be free." (Aquinas 1920, II-II, q. 107, a. 1, ad. 3). When we love someone and will their good, we appreciate that they are the kind of individuals to whom we can will good. Since we cannot love what we do not know, our knowledge of a person's goodness gives rise to our appreciation of the good of the beloved (Pruss 2013, pp. 23–26). When the goodness of individuals includes their generosity to us, gratitude follows. In his *Commentary on Colossians*, Aquinas writes that St. Paul "urges them to acts of love. He mentions two of these acts, peace and thankfulness, and implies a third, joy" (Aquinas 2020c). In this passage, thankfulness is considered an act of love rather than an act annexed to justice. Gratitude to God is therefore related both to justice and to love. In this way, gratitude is like other acts that are related to both justice and love. The act of murder violates love (since the murderer wills what is evil for the victim rather than what is good) but the act of murder is also an act of injustice (since the murder takes what is due to another, the victim's life.) Similarly, in a positive key, gratitude to God, as well as to other persons, involves both justice and love.

Elsewhere, Aquinas writes of gratitude to God as a cause of loving God. In his treatise *On the Two Commandments*, the Dominican says that in order to fulfill the commandment to love God perfectly, four things are required:

The first is the recollection of the divine benefits, because all that we have, whether our soul or body or exterior things, we have them all from God. Therefore, we must serve him with all this and love him with a perfect heart. A man would be extremely ungrateful if, after thinking of all the benefits he received from someone, he did not love him. (Aquinas 1939)

Here, gratitude to God gives rise to the love of God. Thus, for Aquinas, gratitude to God is both a *cause* of loving God and is also an *effect* of loving God. Gratitude to God causes the love of God to increase because when someone thinks of the benefits God has given, that person is prompted to love God more. Likewise, when an individual loves God, the individual is led to appreciate God in terms of giving benefits, which thereby inspires gratitude to God. Gratitude to God causes and is caused by the love of God.

For Aquinas, even adversities are divine gifts. The problem of suffering is, for many people, the most decisive reason to reject God's existence. Can we reconcile the reality of evil with a God of goodness, power, and love? To delve into a theodicy or a defense is beyond the scope of this essay (Stump 2012). But the existence of suffering is, for Aquinas, linked to gratitude to God. In his *Commentary on Ephesians*, Aquinas writes:

The more a person is influenced by his relation to God and knows him, the more does he see God as greater and himself as smaller, indeed almost nothing, in comparison with God. *Now my eye sees you. Therefore do I reprehend myself, and do penance in dust and ashes* (Job 42:5–6). So he declares *giving thanks always for all things*, for all his gifts, whether of prosperity or adversity. *I will bless the Lord at all times; his praise shall be always in my mouth* (Ps 34:1). For adversities are also gifts to us on the way: *count it all joy when you shall fall into diverse temptations* (Jas 1:2). And the apostles *indeed went from the presence of the council, rejoicing that they were*

*accounted worthy to suffer reproach for the name of Jesus* (Acts 5:41). *In all things give thanks* (1 Thess 5:18). (Aquinas 2020d)

Aquinas holds that if we remember the greatness of God and his absolute providential sovereignty over all things, then even temptations should give us joy and our suffering should lead to rejoicing. Aquinas believes that a loving God only permits evil for our own good (Stump 2012). This attitude to suffering is apostolic, but also difficult. In adversity, it is easy to agree with Winston Churchill: "If this is a blessing, it is certainly very well disguised" (Wilson 2006).

To be grateful to God, even in suffering, is also recommended by the example of Job as understood by Thomas Aquinas. This passage comes from *Expositio super Iob ad Litteram*:

> For it would not please God that someone should suffer from adversity unless he wished some good to come to him from it. So though adversity is bitter in itself and generates sadness, nevertheless it should be the cause of rejoicing when one considers the use because of which it pleases God, as is said about the apostles, *the apostles went rejoicing because they had suffered contempt for Christ* (Acts 5:41). For when taking a bitter medicine, one can rejoice with reason because of the hope for health, although he suffers sensibly. So, since joy is the matter of the action of thanksgiving, therefore Job concludes this third argument with an act of thanksgiving, saying, *Blessed be the name of the Lord*. The name of the Lord is truly blessed by men inasmuch as they have knowledge of his goodness; namely, that he distributes all things well and does nothing unjustly. (Aquinas 2020e)

By this time in his story, Job has suffered catastrophic losses, including the death of his children. Aquinas understands the suffering of Job as a severe mercy from God for Job's own well-being. "So though adversity is bitter in itself and generates sadness, nevertheless it should be the cause of rejoicing when one considers the use because of which it pleases God" (Aquinas 2020e). The bitter medicine is awful and there is no denying its terrible power. Rather than stoic detachment, Aquinas recognizes that a good person will feel the deep sadness that comes from catastrophic losses. But Aquinas holds that the bitter medicine is, at the same time, when considered as given by God for our own flourishing, a cause of rejoicing. The chemotherapy both makes us sick and destroys the cancer that is destroying us. Our hope for full recovery of health is a cause for joy.

What does Aquinas mean by saying that "joy is the matter of the action of thanksgiving"? To be joyful is to delight in the good at hand. The good of bitter medicine is not found in the bitterness but in the medical means that enable a good outcome. Even bitter medicine is a gift, not as bitter but as medicine. So, given that we have been given a gift, the gift of hope for future full recovery made possible by means of bitter medicine, we can rejoice even in our sufferings. Joy is the matter of the action of thanksgiving in that what gives joy is the good (or at least what is thought of as good), and it is the good gift that prompts thanksgiving.

In his *Commentary on Colossians,* Aquinas writes,

> So he [St. Paul] says: We thank God, the Author of grace: "Give thanks in all circumstances" (1 Th 5:18). And we thank God always, for the past and for the future. For although we cannot actually pray every minute, we should always pray by serving God out of love: "Pray constantly" (1 Th 5:17); "We ought always to pray" (Lk 18:1)." (Aquinas 2020c)

For Aquinas, prayer need not be limited to vocal prayer (which could not be done every minute of the day). He thinks it is possible to raise the mind and heart to God not just in words but in deeds. Every (morally permissible) action can be done with love of God as its final end. So, everyday activities could, in this view, become ways of thanking God. In his *Commentary on the Psalms*, Aquinas writes, "*I will sing*. Here he [the Psalmist] gives thanks in deed, for to sing is manual labor, and through this a good work is understood, since all our works should end in the glory of God. *So let your light shine before men, that they may see your good works and glorify your Father who is in heaven* (Matt 5:16). *I shall sing*

*to my God as long as I shall be* (Ps 145:2)" (Aquinas 2020a). Not just singing can be a deed giving thanks to God, not just manual labor, but any good work is a way to give thanks to God. In the words attributed to fellow Dominican St. Martin de Porres: "Everything, even sweeping, scraping vegetables, weeding a garden and waiting on the sick could be a prayer if it were offered to God" (Martin 1998, p. 39). As Aquinas puts it, "Every deed, insofar as it is done in God's honor, belongs to religion . . . " (Aquinas 1920, II-II, 81, 1 ad 2). The scope of gratitude to God can include, therefore, anything an agent does knowingly and willingly. All conscious activity can be done in gratitude to God.

## 3. Sins against Gratitude to God

In the *Summa theologiae*, Aquinas proceeds by first introducing a particular virtue, like hope, and then treating deficiencies of the virtue, like despair and presumption. Aquinas views the evil of vice always as a privation of the good of virtue. He thinks we cannot properly understand an evil unless we have some sense of the good in question, which is spoiled by the evil. So, having laid out in the first part of this paper Aquinas' treatment of gratitude to God, we are now situated to consider sins against gratitude to God.

In the *Summa theologiae*, Aquinas indicates that ingratitude comes in three degrees. There can be a failure to recognize the favor bestowed, a failure to expresses thanks for the favor and, finally, a failure to repay the favor in the suitable way (Aquinas 1920, II-II, 107, 2). An agent can fail by omission of any one of these three elements. A still worse kind of failure in gratitude is not to just omit but do the opposite of the three elements of gratitude, namely to view a favor as evil, to find fault with a favor, or to return evil for good. And in so doing, such an individual not only omits what is due in gratitude but does the opposite of what is due in gratitude. These failures might also arise with respect to gratitude to God.

In addition, we can consider what might be called false gratitude to God. Aristotle identified various simulacra of virtues, which look like and may be mistaken for virtues, but in fact are not true virtues. A solider who faces death in battle appears brave, but perhaps he only continues to fight because he is compelled to do so by his commanding officer's threats. The truly brave person is motivated not by threats but for the right reasons (Aristotle 1999). Or, imagine a woman who always drinks alcohol in moderation. She appears to be temperate. But if we learn that she has the mistaken belief that drinking more than two beers is fatal, her "virtue" is in fact not fully virtuous for it is based on ignorance. What appears to be virtue may not in fact be true virtue. So too with gratitude, Aquinas brings to light what might be called "false gratitude".

Consider, for example, the Pharisee who said, "O God, I give you thanks that I am not as the rest of men" (Luke 18:11). The Pharisee gives thanks to God. But is this the virtue of gratitude to God? Aquinas considers the man to be not virtuous but rather haughty. Aquinas says that to be haughty is a form of pride that is exhibited "when one attributes to himself what he has received from another, but considers that he earned it: *I fast twice in a week, I give tithes of all that I possess* (Luke 18:12)." (Aquinas 2020f). The Pharisee appears to be thanking God, but in Aquinas' understanding, the man is attributing his greatness not to God but to himself. He fasts twice a week, so the Pharisee reasons that God must really owe him a lot. The faux grateful person like the Pharisee fails to recognize what he has received from God, and he is haughty because he thinks that his excellence is his own. By contrast, the grateful person recognizes what she has received from God, and so does not attribute to herself what she should not, and so is not haughty in that respect (Konyndyk Deyoung 2004). When considering the poor sinner who is praying, the Pharisee lacks the conviction, saying, "there but for the grace of God, go I." He lacks that realization that his own goodness depends entirely on God.

The case of the proud man who praises God that he is not like other people raises the issue of the relationship between gratitude to God and humility, the opposite of pride. This connection is made clear in one of Aquinas' sermons:

> Imagine, you are a highly placed man or a scholar. You ought to ponder from
> where you have it: you do not have it from yourself, but from God, so that you

may subject yourself to him. And this realization not only takes away pride, but even brings on humility. For as the gifts increase, the reasons for giving honor increase; the more goods you have, the more obliged you are to God. But someone who does not know that the goods he has come from God, cannot thank God. Because of this [I say]: think this over, that whatever you have, you have from God (cf. also Jn 3:27), and that you are bound to give him thanks (cf. Eph 5:20, Col 3:15, et al. *ibi*) or, better, give to him thanks in return. Then those gifts will not lead to pride. (Aquinas 2020g)

Since Aquinas believes that all goods that individuals have come from God, the First Cause, he grounds his vision of human excellence in this reality. To be humble is not to be humiliated, to think that we human beings are mere worms or that we have no value. Rather, Aquinas notes that humility is related to the word for 'ground' (Aquinas 1920, II-II, 61, 1). To be humble is to be grounded in the truth about who we are. We did not give ourselves life. We did not give ourselves our native intelligence. If we have been given great gifts by God, then we owe great gratitude to God. Without gratitude, gifts given by God can lead to pride, and Aquinas believes that pride is deadly for a person's relationship with God and with other people (Aquinas 1920, II-II, 162, 6). So, Aquinas reasons, God might withhold further gifts from those lacking gratitude to God so as not to occasion someone's falling into the sin of pride.

Another manifestation of false gratitude is gratitude for what is mistakenly taken to be a benefit when in fact what is given is not a benefit. Imagine someone who is grateful to another for facilitating sinful behavior. Someone might think it a great gift to be given heroin to use, but this 'gift' entraps the user in deeper addiction. Aquinas writes,

Gratitude regards a favor received: and he that helps another to commit a sin does him not a favor but an injury: and so no thanks are due to him, except perhaps on account of his good will, supposing him to have been deceived, and to have thought to help him in doing good, whereas he helped him to sin. In such a case the repayment due to him is not that he should be helped to commit a sin, because this would be repaying not good but evil, and this is contrary to gratitude. (Aquinas 1920, II-II, 107, 1, ad 1)

Authentic gratitude is for an authentic good. But if a so-called 'gift' is actually harmful, then thankfulness is not called for in this case or, at most, only gratitude for the good will proceeding from mistaken suppositions. To help individuals to sin is to help them in self-harm.

For Aquinas, ingratitude is connected to all sin because all sin involves disobedience to the will of God and a disobedience that manifests an ingratitude to God for his gifts. Indeed, ingratitude might be considered the root of the first sin and the model of every sin. Aquinas writes, "God had conferred on human nature at its beginning, over and above the character of its own principles, that its reason would possess a kind of rectitude of original justice that could impress upon the lower powers without any resistance. And because this had been conferred gratuitously, it was justly taken away through the ingratitude of disobedience" (Aquinas 2020h, II, D.31, a.1). This linking of ingratitude to God and the disobedience of sin is found in another expression in 1542 by St. Ignatius of Loyola who believed ingratitude to be "the cause, beginning, and origin of all evils and sins" (Lehane 2011, p. 16).

Yet, is not at least the original sin traditionally linked with pride rather than ingratitude? Yes, but for Aquinas ingratitude is itself linked to pride: "Disobedience and ingratitude are not species of pride as if dividing pride essentially but are called species of pride as possessing a certain participation in pride, insofar as they are commanded by pride" (Aquinas 2020h, II.D42.Q2.A4.C.5). The ungrateful person has a distorted view of his own excellence as self-caused and not ultimately due to God's gift. For this reason, Aquinas writes that the sin of a more excellent person is aggravated in comparison to the sin of a less excellent person "on account of ingratitude, because every good in which a man

excels, is a gift of God, to Whom man is ungrateful when he sins: and in this respect any excellence, even in temporal goods, aggravates a sin, according to Wis. 6:7: *The mighty shall be mightily tormented*" (Aquinas 1920, I-II, 73, 10). The greater the gifts we have received, the greater the ingratitude of disobedience.

This understanding of ingratitude to God leads Aquinas to reason that God may withhold some gifts from ungrateful people, knowing from all eternity that if certain gifts were given to individuals, if the individuals were thereby to become more excellent than they are, these gifts would aggravate their sin (Aquinas 2022n, C1, L5, n.75). God sometimes withholds giving good gifts to an ungrateful individual since "not undeservedly, did condemnation follow his ingratitude for that same good. And owing to that ingratitude, what is good became evil to him, as happens to them who receive Christ's body unworthily" (Aquinas 1920, III.81.2). God's mercy is sometimes expressed through withholding gifts to the ungrateful, since these gifts, despite being good in themselves, could inadvertently harm the ungrateful.

For Aquinas, sin is the worst kind of evil that a human person can suffer. If by not giving further gifts God can prevent someone from falling into further alienation from God, it is a mercy, a severe mercy, for God not to give that person gifts. By analogy, imagine good parents withholding a cash gift to an addicted daughter because they foresee that she will actually be made worse off having more money to buy more drugs. God, foreseeing that a gift will be an occasion for the sin of ingratitude and perhaps other sins as well, could judge that people are better off, all things considered, not receiving a possible gift.

On the other hand, to give thanks to God is to occasion even greater blessings. Aquinas writes, "to the source whence blessings come they return, namely, by giving thanks, to flow again by repeated blessings" (Aquinas 2022n, C1.L5.n75.3). Aquinas does not specify what these blessings are, but given what he says elsewhere, to return thanks to God is itself a gift: the gift of intensifying the relationship to God. In giving thanks to God, we receive the blessing of turning our minds to God, of acknowledging his goodness in giving gifts to us, and of recognizing God's love for us. When we give thanks to God, it becomes easier to know and love God, which is, for Aquinas, the ultimate end of the human agent.

## 4. Jesus and Gratitude to God

In his book *Gratitude: An Intellectual History,* Peter Leithart offers one of the only works, and certainly the most influential work, on the history of gratitude. Because he is a prolific author, minister, and theologian, Liethart's voice is particularly an influential one in the Christian community, particularly when the topic is gratitude. Leithart writes:

> Little of Aquinas' account is distinctively Christian. Believing as he does in creation, he recognizes that all things are gifts of God. He, of course, endorses the Pauline exhortation to "give thanks in all circumstances. Yet when he gives direct attention to gratitude, he follows Seneca and Tully to give a slightly Christianized version of ancient reciprocity. In his work the infinite circle of Christian gift and gratitude contracted, and this contraction was perpetuated into the following centuries. (Leithart 2014, p. 94)

Liethart's observations are fairly accurate as far as the *Summa theologiae* goes, but are less accurate in terms of Aquinas' opera omnia, particularly his Biblical commentaries. Jesus does make a difference for Aquinas' understanding of gratitude.

For Aquinas, proper gratitude to God is incomplete without a knowledge of Christ. Aquinas writes:

> All the knowledge imparted by faith revolves around these two points, the divinity of the Trinity and the humanity of Christ. This should cause us no surprise, for the humanity of Christ is the way by which we come to the divinity. Therefore, while we are still wayfarers, we ought to know the way leading to our goal. In the heavenly fatherland adequate thanks would not be rendered to God

if men had no knowledge of the way by which they are saved. (Aquinas 2020m, ch.2)

We cannot be grateful if we do not know the favor we have received. Aquinas believes God saved us through the gift of God himself, who was made man, born of Mary, suffered, died, and rose from the dead for us. Thus, adequate gratitude to God, and giving thanks is part of this adequate gratitude, requires knowledge of Jesus. So, who is Jesus?

For Aquinas, Jesus was and is a perfect human being as well as perfect God. If Jesus is God, this complicates gratitude to God. Unlike God the Father who (according to Aquinas) does not suffer in giving us gifts, Aquinas believes that Jesus did suffer in giving us the gift of salvation by taking on human nature, suffering on the cross, and dying on behalf of sinners (Aquinas 1920, III, 46, 6). According to some accounts of gratitude, when the giver suffers in giving us a gift, we owe the giver a greater debt of gratitude than when the giver gives the gift without personal cost. If this principle is correct, the Christian owes a greater debt of gratitude to God than the kind of theist who holds that God does not suffer in giving us gifts.

Aquinas argues that Jesus taught us about gratitude to God in teaching his disciples how to pray. The Lord's Prayer, the Our Father, helps us to avoid the sin of ingratitude. Aquinas writes, "In these very words [give us this day our daily bread] the Holy Spirit teaches us to avoid five sins which are usually committed out of the desire for temporal things" (Aquinas 2020i, S4.3). Aquinas goes on to say:

> The fifth sin is ingratitude. A person grows proud in his riches, and does not realize that what he has comes from God. This is a grave fault, for all things that we have, be they spiritual or temporal, are from God: *all things are thine; and we have given thee what we received of thy hand* (1 Chr 29:14). Therefore, to take away this vice, the prayer has, *give us,* and *our bread,* that we may know that all things come from God. (Aquinas 2020i, S4.8)

Aquinas, in other words, thinks that the very prayer given to Christians by Jesus contains within it a teaching of the Holy Spirit about avoiding the sin of ingratitude.

Was Jesus an ingrate? A discussion of gratitude and the role of Jesus would be incomplete without a consideration of Liethart's claim that "Jesus was an ingrate" (Leithart 2014, p. 68). Although Jesus gave gratitude to God, Liethart holds that nowhere in the Gospels is it recorded that Jesus gave thanks to any human being. Would Aquinas agree with this reading of the Bible?

Having searched through Aquinas' commentaries on Scripture, both the free-standing commentaries and the interpretations of Biblical passages about Christ in texts such as the *Summa theologiae* and the *Summa contra Gentiles*, I found no passage contradicting Liethart's claim that Jesus never explicitly said "thank you" to any human being. Thus far, Aquinas and Liethart agree.

However, Aquinas argues, echoing John 21:25, that "Our Lord did and said many things which are not related in the Gospel" (Aquinas 1920, IV.29.3 reply to 1). There is no explicit passage in Scripture that Jesus consumed any nourishment between Mary's nursing him as a baby and his public ministry as a full-grown adult. But if Jesus was fully human, then he obviously ate food during this roughly thirty-year period between being an infant and being an adult in public ministry. There is no passage that talks about Jesus as a five-year-old or as a twenty-year-old, but Jesus must have spent time at these ages. So, the lack of Biblical passages in which Jesus shows gratitude to human beings would not be, for Aquinas, an indication that Jesus in fact never thanked other people.

Aquinas thinks that there is a good reason that Scripture does not record all the actions of Jesus. Aquinas writes:

> For to write about each and every word and deed of Christ is to reveal the power of every word and deed. Now the words and deeds of Christ are also those of God. Thus, if one tried to write and tell of the nature of every one, he could not do so; indeed, the entire world could not do this. This is because even an infinite

number of human words cannot equal one word of God. From the beginning of the Church, Christ has been written about; but this is still not equal to the subject. Indeed, even if the world lasted a hundred thousand years, and books written about Christ, his words and deeds could not be completely revealed. (Aquinas 2020j, C21.L6.n.2660)

As Scripture itself indicates, Scripture does not exhaust or capture in their fullness all the words and deeds of God in Christ, "But there are also many other things which Jesus did; which, if every one of them were to be written, the world itself, I think, would not be able to contain the books that should be written" (John 21:25). So, we should not reason from the silence of Scripture to the conclusion that Jesus did *not* give thanks to human beings.

For Aquinas, there is reason to think that Jesus did in fact give thanks. Jesus perfectly fulfills the laws of the Old Testament (Matt 5:17). One of the fundamental commands of the Old Testament is to "honor your father and mother" (Exodus 20:12). So, Jesus must have perfectly honored both his father and mother. Now, for Aquinas at least, honoring parents is one manifestation of gratitude (broadly speaking, since properly speaking the virtue of responding properly to the gifts of parents is 'piety'.) Recall that Aquinas differentiates various kinds of debts: "Corresponding to these various kinds of debt there are various virtues: e.g., *Religion* whereby we pay our debt to God; *Piety*, whereby we pay our debt to our parents or to our country; *Gratitude*, whereby we pay our debt to our benefactors, and so forth" (Aquinas 1920, I-II, 60, 3). Since Christ perfectly fulfilled Jewish law (Matthew 5:17–20), Jesus must have shown gratitude to his mother Mary (Hahn 2006).

Even if it is true that Jesus showed gratitude for his mother, was Jesus an ingrate to other human beings? To consider how Aquinas would answer this question, recall that Aquinas looks to Jesus as the human paradigm and model for how to live: "In His manner of living our Lord gave an example of perfection as to all those things which of themselves relate to salvation" (Aquinas 1920, ST III, 40, 2, Reply to 1). In his essay, "Jesus in the Moral Theology of Thomas Aquinas," Joseph Wawrykow notes, "Jesus is the model for authentic behavior, the great human exemplar who shows what is possible for those who are in correct relationship to God, who indicates in his own action how they might act as they move toward God as their end" (Wawrykow 2012, p. 21). According to Aquinas, Jesus had all the virtues: "Now the more perfect a principle is, the more it impresses its effects. Hence, since the grace of Christ was most perfect, there flowed from it, in consequence, the virtues which perfect the several powers of the soul for all the soul's acts; and thus Christ had all the virtues" (Aquinas 1920, III, 7, 2). Christ not only had the virtues, "He had them most perfectly beyond the common mode" (Aquinas 1920, III, 7, 2, ad 2). The perfect human being was perfect in virtue.

If Christ has every virtue, and if Aquinas is right that gratitude is a virtue, then Christ must have had gratitude to the people who helped him. There were many. Mary helped Jesus by carrying him in her womb, giving birth to him, and caring for him as a baby. Joseph helped baby Jesus by finding a place for him to be born, by protecting him from King Herod who sought to kill him, and by taking Jesus safely to Egypt and out of Egypt. The magi brought gifts of gold, frankincense, and myrrh to baby Jesus. His cousin John helped Jesus by announcing the Lamb of God to the world. Peter and John helped Jesus to arrange the place to celebrate the Last Supper. Simon of Cyrene helped Christ carry his cross. This list is not exhaustive. If Jesus has the perfection of every virtue, and if Aquinas is right that gratitude is a virtue, then Christ must have had gratitude to these people who helped him.

Indeed, although the words "thank you" are not used, Jesus does exhibit gratitude to the woman who anoints him with expensive perfume, an event recorded in all four Gospels (Matthew 26, Mark 14, Luke 7, and John 12). As found in the Gospel of Matthew:

> While Jesus was in Bethany in the home of Simon the Leper, a woman came to him with an alabaster jar of very expensive perfume, which she poured on his head as he was reclining at the table. When the disciples saw this, they were

indignant. "Why this waste?" they asked. "This perfume could have been sold at a high price and the money given to the poor." Aware of this, Jesus said to them, "Why are you bothering this woman? She has done a beautiful thing to me. The poor you will always have with you, but you will not always have me. When she poured this perfume on my body, she did it to prepare me for burial. Truly I tell you, wherever this gospel is preached throughout the world, what she has done will also be told, in memory of her." (Matthew 26:6–13)

Is it fair to construe Jesus as expressing gratitude in this passage? The answer to that question depends in part on how gratitude is defined. We could define gratitude as Aquinas does, in terms of "[1] recollecting the friendship and kindliness shown by others, and [2] in desiring to pay them back . . . ." (Aquinas 1920, II-II, 80, 1). Or, we might follow Robert Emmons, who defines gratitude as "a willingness to recognize (a) that one has been the beneficiary of someone's kindness, (b) that the benefactor has intentionally provided a benefit, often incurring some personal cost and (c) that the benefit has value in the eyes of the beneficiary" (Emmons 2013, p. 5).

If we define gratitude according to either of these definitions, Jesus shows gratitude to the woman. Jesus recognizes that he has been the beneficiary of someone's kindness: "She has done a beautiful thing to me." He sees that she has intentionally provided a benefit incurring personal cost in using "very expensive perfume." Finally, this exuberant gift has value in the eyes of Jesus: "When she poured this perfume on my body, she did it to prepare me for burial." Jesus desires to and does return the favor immediately by defending the woman from criticism: "Why are you bothering this woman?" Christ praises her kind deed and glorifies the woman by saying, "Truly I tell you, wherever this gospel is preached throughout the world, what she has done will also be told, in memory of her." If we understand gratitude as Aquinas and Emmons define it, then Jesus was not an ingrate to this woman.

We have no reason to think that the example of this woman or these other examples of those giving to Jesus mentioned earlier definitively lists all the gifts Jesus was given. Indeed, the number of people who helped Jesus includes all those who helped those who helped Jesus. In as much as the Body of Christ is extended through time, found both in the Church and in those in need (Matthew 25), those who have served Jesus include a vast multitude from every nation, tribe, people, and language (Rev. 7:9).

So, since Jesus received kindness and favors by others many times, there are grounds for Jesus to be grateful to others. It would be, then, a failure on the part of Jesus if he were not grateful. If it is true that gratitude is a virtue (gratitude not just to God but to other human beings), and if Jesus had all the virtues, then Jesus had the virtue of gratitude, as well as piety, and religion. In other words, Jesus was not an ingrate.

We can approach the question of whether Jesus was an ingrate in another way. Aquinas views Jesus as living a perfectly blameless life: "Christ wished to make His Godhead known through His human nature. And therefore, since it is proper to man to do so, He associated with men, at the same time manifesting His Godhead to all, by preaching and working miracles, and by leading among men a blameless and righteous life" (Aquinas 1920, III, 40, 1, Reply to 1). Aquinas thinks that Jesus is not only sinless but that "Christ was incapable of sin" (Aquinas 1920, III, 40, 3, ad 1). So, if it is true that ingratitude is a sin, then Jesus not only did not commit any sins of ingratitude, but could not commit any sins of ingratitude. Jesus was not an ingrate.

There are at least two possible responses to these considerations. First, it could be denied that gratitude is a virtue or that ingratitude is a vice. Although Liethart might disagree, Aquinas holds that gratitude is a virtue and ingratitude is a vice, so this way of defending the conclusion that "Jesus is an ingrate" is not available to Aquinas.

Second, it might be admitted that gratitude is virtuous and ingratitude sinful, but Jesus might (for whatever reason) not be subject to these categories. For example, although Aquinas holds that Jesus had the fullness of all the virtues, Aquinas qualifies this generalization later when he writes that Jesus did not have the virtues of faith or of hope.

For Aquinas, Jesus from the beginning of his human existence enjoyed the beatific vision (Aquinas 1920, III, 15, 10). And for Aquinas, faith is not had by those who enjoy the beatific vision for they see God face to face without faith. Likewise, hope for *future* eternal life does not exist in those who *now* enjoy eternal life in the beatific vision. So, Aquinas concludes that Jesus had neither faith nor hope, thus qualifying the claim that Jesus had all the virtues. So, perhaps, also with gratitude to human beings, Jesus did not have it even though (for other people) gratitude is a virtue and ingratitude is a vice.

Is gratitude like faith and hope, a virtue needed in general by human beings but not needed by Jesus? Recall that the reason Aquinas gives that Jesus does not have faith and hope is that Jesus was not just a wayfarer who journeyed towards God, but a comprehensor who enjoyed the beatific vision of God (Aquinas 1920, III, 15, 10). As such, he did not have faith or hope but saw and enjoyed God (Aquinas 1920, III, 7, 3 and III, 7, 4). But gratitude to God, as Aquinas noted earlier, continues in heaven (Aquinas 1920, IV Q16, a.2). Does gratitude also extend to human beings in heaven? If it does, then the beatific vision enjoyed by Christ is no obstacle to Jesus expressing gratitude to human beings.

Aquinas does not, as far as I can tell, ask and answer this question explicitly. But in the parable of the talents, Jesus teaches an eschatological story replete with gratitude: "His master replied, 'Well done, good and faithful servant! You have been faithful with a few things; I will put you in charge of many things. Enter into the joy of your master!'" (Matthew 25:23). In Aquinas' interpretation, the master recognizes the good that the servant has done and rewards him for it (Aquinas 2020k, C25.L2.n2052.4). Although the word "gratitude" does not appear in this passage, the words of the master include an acknowledgement of the good done on behalf of the master, praise for this work, and a repayment of sorts—classic elements of gratitude. If the master of the parable represents Christ the Divine Master, then Jesus himself has gratitude to his servants. Indeed, for Aquinas, the faithful servant is a friend of the master:

> The master ought to feel towards his servant as a friend, hence it is said, *as your own soul*. For this is proper to friends, that they are of one mind in what they will and what they do not will. *Now the multitude of the believers were of one heart and one soul* (Acts 4:32). By which we are given to understand that there is a consensus of master and servant, when the faithful servant becomes a friend. (Aquinas 2020l, p. 2)

In the Gospel of John, Jesus calls his disciples not only servants but friends (John 15:15). Indeed, Aquinas says that "Christ is our wisest and greatest friend" (Aquinas 1920, I-II, 108, 4, also see Ryan 2016). If followers of Jesus are also friends of Jesus, then the gratitude of Christ to human beings is even more appropriate, since friends show one another thanks for gifts received. And this friendship (charity) between God and human beings does not end, but finds its completion in heaven (Aquinas 1920, II-II, 24, 8). So, although Aquinas does not explicitly address the question of whether Jesus has gratitude to human beings in heaven, this conclusion would not only be compatible but would also seem to follow from the parable of the faithful servants and the friendship that exists in the life to come.

Although it is surprising that there is no explicit passage recording Jesus thanking a human being, Aquinas would not conclude from this lacuna that Jesus was an ingrate. If gratitude is a virtue, and if Jesus had all the virtues, then Jesus had gratitude. In all four Gospels, Jesus expresses classic elements of gratitude to the woman who anoints him with expensive perfume. In the parable of the talents, the Good Master (Christ) acts with gratitude to the faithful servants who invested their talents and made a return. Jesus was no ingrate.

## 5. Conclusions

This paper has attempted to bring together various passages from Aquinas' *opera omnia* to shed light on his conception of gratitude to God and ingratitude to God, as well as the transformation Jesus makes to gratitude to God. In taking into account the totality of Aquinas' written work, the full scope of his views of gratitude to God move it beyond a

simplistic application of Stoic ideas of gratitude. This essay also argues that Jesus was no ingrate both because of how he treats the woman who anoints him and because Jesus had the fullness of all virtues, including the virtue of gratitude.

**Funding:** This research was funded by the Templeton Foundation, grant number 59916.

**Institutional Review Board Statement:** Not applicable.

**Informed Consent Statement:** Not applicable.

**Data Availability Statement:** Not applicable.

**Conflicts of Interest:** The author declares no conflict of interest.

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
