# Peer review of "Thomas Aquinas on Gratitude to God"

_religions, doi:10.3390/rel13080692_

Round 1

Reviewer 1 Report

I enjoyed reading the paper--it's informative and a helpful guide to Aquinas's view on gratitude.  I am pleased to recommend publication.  I have only a handful of minor comments, as follows:

(1) typo on line 43  "Accordingly..."

(2) awkward construction of introduction to quoted material on line 53.

(3) than not that at line 57

(4) lines 126-143:  Seems to me that you need to introduce the distinction between acquired and infused virtue first.  As it stands, the paragraph where you introduce Aquinas's definition of virtue seems odd until that distinction is introduced.  Might considering re-ordering things for lucidity.

(5)  Sometimes you say "Thomas" and sometimes you say "Aquinas" and sometimes "Thomas Aquinas".  Consistency?

(6) The argument you make at lines 160-165 seems a bit uncompelling to me, though I don't quite know why to be honest.  Is there something else that could be batted around here to explain and/or bolster the argument at play?

(7) extra space on line 165

(8) lines 176-180--Similar problem as above.  This just doesn't seems fully compelling.  Perhaps gratitude is needed for FULL or ROBUST expressions of the same virtues?

(9) lines 208-216:  I confess that I don't quite see how this quotation illustrates the point at hand.  Perhaps a bit more reflection on this quote would help.

(10) line 221:  need comma after Summa theologiae

(11) line 235-236.  Commas after God and after persons

(12) lines 235.  cut extra space between paragraphs

(13).  line 258:  Odd citation to Stump here.  Seems unnecessary.

(14) line 278:  Sometimes you cite ENGLISH and sometimes you cite LATIN titles of Aquinas.  Consistency?

(15) line 319:  prayer is, not prayer as

(16) line 327:  singing not signing?

(17) line 390:  God, the First Cause.   (Has the comma just fallen out of vogue these days?)  ;)

(18) line 415:  too many instances of the word gratitude.  

(19) lines 430-433:  please check original.  Something about the sentence seems off to me after "essentially"

(20) You misspell Liethart's name in the essay but not the bib.  Fix every instance.

(21)  I'm not quite sure I understand why you have selected Leithart as your foil in section 4.  Is it just that you want to correct the record?  Fair enough.  But perhaps you should tell the reader why you are doing that.  As it stands, a discussion of Liethart seems a bit un-motivated to the reader,  or at least this reader.  (And, frankly, I like Liethart, so that's not the problem here.)  Seems to me, though, that you should maybe tell the reader why you are ending this essay with your entanglement with Liethart.  Is it because he has an important work on gratitude?  Something needed.

(22) line 534:  should "related" be "relayed"?  Check original.

(23)  around line 540:  You're right here.  This seems pretty obvious to me, which raises the question of "Why Liethart in the first place?"  Arguments from biblical silence seem to be in a special class of terrible argument.  Per the biblical record, Jesus had nothing to say about lots of stuff, including instrumental music in worship.  It does not follow that...

(24) Lines 718ff.  Look over your bib.  City of publication missing on many entries.  Also the order of the listing of city of pub and publisher seems off on several entries.  

Overall, solid work, and I look forward to seeing the final published article.

Author Response

Thank you for this careful review. Please find my responses below.

(1) typo on line 43  "Accordingly..."

Fixed. 

(2) awkward construction of introduction to quoted material on line 53.

Yes, I’ve tried now  to put my point more clearly, “So, gratitude as a matter of disposition rather than a deed. This distinguishes gratitude from justice in typical cases. The just act must be accomplished, rather than simply willed. By contrast, Thomas thinks of gratitude as primarily a matter of will, rather than of external deeds of repayment.”

(3) than not that at line 57

Corrected. 

(4) lines 126-143:  Seems to me that you need to introduce the distinction between acquired and infused virtue first.  As it stands, the paragraph where you introduce Aquinas's definition of virtue seems odd until that distinction is introduced.  Might considering re-ordering things for lucidity.

Good point. Now the text reads, “Thomas Aquinas distinguishes two kinds of virtue. The infused virtues we can through a free gift of God. The acquired virtues are gained through our own repeated actions. In defining what an infused virtue is, Thomas endorses the definition he finds in Augustine of Hippo,...”

(5)  Sometimes you say "Thomas" and sometimes you say "Aquinas" and sometimes "Thomas Aquinas".  Consistency?

In scholarship on Aquinas, his name is repeated so often that sometimes he is called “Thomas” other times “Aquinas” and still other times “Thomas Aquinas” just for the sake of variety, lest the repetition of the name in writing be too monotonous.  

(6) The argument you make at lines 160-165 seems a bit uncompelling to me, though I don't quite know why to be honest.  Is there something else that could be batted around here to explain and/or bolster the argument at play?

I’m not sure what else to add beyond what I’ve said, so I’ll leave what I’ve said unaltered. 

(7) extra space on line 165

Extra space is now removed.

(8) lines 176-180--Similar problem as above.  This just doesn't seems fully compelling.  Perhaps gratitude is needed for FULL or ROBUST expressions of the same virtues?

I’m not sure what else to add beyond what I’ve said, so I’ll leave what I’ve said again unaltered. 

(9) lines 208-216:  I confess that I don't quite see how this quotation illustrates the point at hand.  Perhaps a bit more reflection on this quote would help.

I’ve now added, “If we consider a virtue to be greater inasmuch as it is more long lasting, then gratitude to God is greater in this respect than faith or hope, both of which no longer exist in heaven. So, according to Thomas, in heaven we will both remember our sins and give thanks to God for his mercy in forgiving our sins which makes gratitude to God, in this respect, greater than faith and hope.”

(10) line 221:  need comma after Summa theologiae

Done.

(11) line 235-236.  Commas after God and after persons.

Done.

(12) lines 235.  cut extra space between paragraphs

It is gone now.

(13).  line 258:  Odd citation to Stump here.  Seems unnecessary.  

I agree it is not necessary to cite Stump here, but I’d like to make reference to this book in case someone needs to understand the difference between theodicy and defense.

(14) line 278:  Sometimes you cite ENGLISH and sometimes you cite LATIN titles of Aquinas.  Consistency?

It is common in the scholarship on Aquinas to cite English or Latin without bothering to always have one or the other. 

(15) line 319:  prayer is, not prayer as

Fixed.

(16) line 327:  singing not signing?

Yes, singing, not signing.

(17) line 390:  God, the First Cause.   (Has the comma just fallen out of vogue these days?)  ;)

Comma now, added. ;)

(18) line 415:  too many instances of the word gratitude.  

This mistake has now been corrected.

(19) lines 430-433:  please check original.  Something about the sentence seems off to me after "essentially"  

I did check the original and the text is correct as written. 

(20) You misspell Liethart's name in the essay but not the bib.  Fix every instance.  

I’ve now corrected all the mispellings. 

(21)  I'm not quite sure I understand why you have selected Leithart as your foil in section 4.  Is it just that you want to correct the record?  Fair enough.  But perhaps you should tell the reader why you are doing that.  As it stands, a discussion of Liethart seems a bit un-motivated to the reader,  or at least this reader.  (And, frankly, I like Liethart, so that's not the problem here.)  Seems to me, though, that you should maybe tell the reader why you are ending this essay with your entanglement with Liethart.  Is it because he has an important work on gratitude?  Something needed.

I’ve added a bit in order to respond to this concern. The text now says, “In this book Gratitude: An Intellectual History, Peter Leithart offers one of the only and certainly the most influential books on the history of gratitude. Because he is a prolific author, minister, and theologian, Liethart’s voice is particularly an influential one in the Christian community, particularly when the topic is gratitude.” 

(22) line 534:  should "related" be "relayed"?  Check original. It is “related” not “relayed” (though relayed also makes sense).

(23)  around line 540:  You're right here.  This seems pretty obvious to me, which raises the question of "Why Liethart in the first place?"  Arguments from biblical silence seem to be in a special class of terrible argument.  Per the biblical record, Jesus had nothing to say about lots of stuff, including instrumental music in worship.  It does not follow that…

Yes, I agree the argument from silence is not a strong one. I’ve engaged with Leithart both because he is influential and because it is worth setting the record straight. 

(24) Lines 718ff.  Look over your bib.  City of publication missing on many entries.  Also the order of the listing of city of pub and publisher seems off on several entries.  

I’ve corrected the missing cities of publication. 

Reviewer 2 Report

I really enjoyed this paper, even though it is somewhat outside my wheelhouse. (I'm not a philosopher or a theologian.) It gave me a lot of useful ideas that I hope to be able to explore further in my own work. 

My only substantive question is whether the work on whether Jesus is an ingrate is the best fit for this article, since it was mostly focused on interpersonal gratitude by Jesus vs. gratitude to God in the rest of the piece. To me it broke the flow and felt like something different that might merit a short, stand-alone article elsewhere. But I was OK with it. 

My other comments are all minor editing things. 

--I was surprised to see "Thomas" referenced throughout the paper, as opposed to using "Aquinas." Perhaps this is standard in current theological or philosophical writing, but to me it seemed curious. 

--Typos: 

  --line 3 "the gratitude to God"

   --57:  THAN the deed of repaying?

  -- 66: "chiefly on the heart"?

   ---319: "prayer was not"?

   ---327: "not just signing"  --> singing?

  --434: "to due"  --> "due to"?

  --435: sin of a less excellent person?

   --

Author Response

Thank you for your review. I do see the point that the “Jesus as ingrate” part of the essay could be its own stand alone (short) essay. But I do think it is important to have it in my essay both because of the role of Jesus in Thomas’s understanding of God and because Leithart’s work is so important and influential. I think if I did not include at least something about Leithart’s work scholars would wonder why he was omitted. 

I’ve also responded to all minor editing things. 

--I was surprised to see "Thomas" referenced throughout the paper, as opposed to using "Aquinas." Perhaps this is standard in current theological or philosophical writing, but to me it seemed curious. 

In scholarship on Aquinas, his name is repeated so often that sometimes he is called “Thomas” other times “Aquinas” and still other times “Thomas Aquinas” just for the sake of variety, lest the repetition of the name in writing be too monotonous.  

--Typos: 

  --line 3 "the gratitude to God"

Fixed.

   --57:  THAN the deed of repaying?

Fixed.

  -- 66: "chiefly on the heart"?

Fixed.

   ---319: "prayer was not"?

Changed to, “For Thomas, prayer need not be limited to vocal prayer….”

   ---327: "not just signing"  --> singing?

Fixed.

  --434: "to due"  --> "due to"?

Fixed.

  --435: sin of a less excellent person?

  Fixed.